# A Nano Drug Delivery System Based on *Angelica sinensis* Polysaccharide for Combination of Chemotherapy and Immunotherapy

**DOI:** 10.3390/molecules25133096

**Published:** 2020-07-07

**Authors:** Min-Zhe Wang, Xin He, Zhe Yu, Hong Wu, Tie-Hong Yang

**Affiliations:** Department of Medicine Chemistry and Pharmaceutical Analysis, School of Pharmacy, Air Force Medical University, Xi’an 710032, China; wmz513yy@163.com (M.-Z.W.); hexin960229@163.com (X.H.); yzfmmu@163.com (Z.Y.)

**Keywords:** *Angelica sinensis* polysaccharide, doxorubicin, enzymes sensitive drug delivery, synergistic therapy

## Abstract

Combination of chemotherapy and immunotherapy has been a promising strategy in cancer treatment. Polysaccharides from *Angelica sinensis* (AP), a well-known Chinese herbal medicine, have been proved to have good immunomodulatory activity. In the present study, an enzyme-sensitive tumor-targeting nano drug delivery system (AP-PP-DOX (doxorubicin), PP stood for peptide) was constructed. In this system, *Angelica* polysaccharides act as not only carriers to targeted delivery of drugs to tumor tissue but also effectors to improve tumor microenvironment and enhance immune function, resulting in synergistic antitumor effect with chemotherapy drugs. The structure of this conjugate was confirmed by FI-IR and ^1^H-NMR. The particle size and zeta potential of the nanoparticles were 129.00 ± 3.32 nm and −28.45 ± 0.22 mV, respectively. Doxorubicin (DOX) and AP could be quickly released from the AP-PP-DOX under the presence of matrix metalloproteinase 2 (MMP2). The released DOX showed good antitumor efficacy *in vitro*. The treatment of released AP moiety increased the expression of IL-2, while that of IL-10 was decreased, showing potential in restoring Th1/Th2 immune balance in tumor microenvironment. In a word, this drug delivery system, with specific tissue targeting and tumor microenvironment improvement, will open a new avenue for combination treatment of cancer.

## 1. Introduction

Lack of tumor cell selectivity and abnormal tumor immune microenvironment are two key factors resulting in low chemotherapeutic efficacy. Combination of traditional chemotherapy for tumor cells and improvement of tumor microenvironment can provide a more effective and safer synergistic strategy for clinical intervention of tumor.

It has been shown in recent studies that polysaccharides from traditional Chinese medicine (TCM) can improve the immunosuppressive state of tumor microenvironment to play an antitumor effect [1], through regulating T lymphocyte subsets in the tumor microenvironment, promoting the maturation of dendritic cells (DC), activating macrophages, and inhibiting tumor angiogenesis. In our previous researches, it was found that *Angelica* polysaccharide (AP) could inhibit tumor growth and enhance antitumor immunity [2,3,4,5,6].

Nano drug delivery systems have been widely used in target-specific drug delivery systems and showed advantages in cancer treatment. Natural polysaccharides have been widely used in nano drug delivery system [7,8]. However, researches so far only used polysaccharides as carrier materials and neglected their possible biological activities. In the present study, an enzyme-sensitive tumor-targeting nano drug delivery system (AP-PP (peptide)-DOX (doxorubicin)) was constructed, intended dual functions of killing tumor cells and improving tumor microenvironment. In this system, *Angelica* polysaccharides act as not only carriers to targeted delivery of drugs to tumor tissue but also effector to improve tumor microenvironment and enhance immune function, resulting in synergistic antitumor effect with a chemotherapy drug (Figure 1).

## 2. Results and Discussion

### 2.1. Construction and Characterization of AP-PP-DOX Conjugates

The AP-PP-DOX conjugates were synthesized as shown in Figure 2. It is reported that some peptide sequences such as GPLGIAGQ, GPLGV, and PLGLAG have been shown matrix metalloproteinase 2 (MMP2)-cleavable character [9,10]. In this study, NH_2_-GPLGIAGQC-SH was designed as the MMP2 cleavable peptides (PP).

AP was first decorated with maleic anhydride (MA) through an ester bond to generate AP-MA. The product was verified by FT-IR spectra shown in Figure 3a. A very broad peak from 2900 to 3500 cm^−1^ was attributed to the -OH absorption of the carboxylic acid in the MA segment and the unreacted hydroxyl groups of AP. The presence of a carbonyl peak at 1735 cm^−1^ in AP-MA was evident due to the ester linkage, as well as the carboxylic acid end group. The ^1^H-NMR spectra of AP-MA showed several distinctive peaks in the double bond region (6.2–6.5 ppm), which were not present in the original AP (Figure 3b).

AP-PP was synthesized by a reaction between carboxyl group of AP-MA and amino group of C-terminal amino acid of the PP peptide. The ^1^H-NMR spectrum of AP-PP in D_2_O was shown in Figure 3b. Peaks between 4.25 and 4.50 ppm were a characteristic shift representing -NCHCO-. The -CH_3_ of PP was seen between 0.8 and 1.2 ppm. Furthermore, the characteristic peaks of AP were also observed between 3.5 and 4.0 ppm. The ^1^H-NMR results indicated that the compound was in agreement with the predicted structure of AP-PP.

To conjugate DOX to the AP-PP, DOX was first maleimidated by the amine-to-sulfhydryl crosslinker, 3-(Maleimido) propionic acid *N*-hydroxysuccinimide ester (SMP). The DOX-SMP was synthesized by a reaction between the 3-amino group of the daunosamine sugar of DOX and an active ester group of SMP. Triethylamine (TEA) was added to remove the hydrochloride salt and also to maintain basic conditions to favorably generate DOX-SMP. The FT-IR spectra of DOX-SMP were shown in Figure 3c. The characteristic peaks at 3394 cm^−1^ and 1639 cm^−1^ were attributed to the stretching vibration of -NH- and C=O in amide Ι band, respectively. The appearance of a secondary amide showed that DOX and SMP were coupled by an amide linkage.

The *C*-terminal amino acid of the PP peptide is a cysteine, which provides a chemically reactive thiol group for drug conjugation. DOX-SMP can be covalently attached to the cysteine sulfur of PP with maleimide by way of a Michael addition (nucleophilic addition) to produce AP-PP-DOX. In the ^1^H-NMR spectra of AP-PP-DOX (Figure 3d), the AP was characterized by the peak of -OH protons (3.64 ppm), the PP was characterized by the peak of -CH_3_- protons (0.91 ppm) and -NCHCO- protons (4.25–4.50 ppm), and the DOX was characterized by the peak of anthracene ring protons (7.78 ppm).

Thus, according to the well-designed synthetic route and proven by the FT-IR and ^1^H-NMR spectrum result, a derivative of DOX linked with AP (AP-PP-DOX) was obtained successfully.

### 2.2. Nanoparticle Formation and Drug Loading

As an amphiphilic polymer, AP-PP-DOX could form a nanoparticle by self-assembly rather than be fully dissolved in PBS [11]. The hydrophobic DOX forms the “core” of the nanoparticles, and the hydrophilic AP forms the “shell” of the nanoparticles, which was evidenced by their particle size of 129.00 ± 3.32 nm (Figure 4a) and the polydispersity index was 0.119. The zeta potential of the nanoparticles was −28.45 ± 0.22 mV (Figure 4b). The morphology of the nanoparticles, which was analyzed by transmission electron microscopy (TEM), also confirmed the formation of AP-PP-DOX nanoparticles. The nanoparticles appeared as circular and uniform (Figure 4c). The particle size of the nanoparticles was about 100 nm in TEM analysis, which was smaller than that in dynamic light scattering (DLS) analysis. This is because the nanoparticles were detected in solution in DLS method while the sample was dry in TEM analysis.

The DOX drug loading efficiency (DL) of the nanoparticles was also measured by ultraviolet spectrophotometer after the nanoparticles were decomposed. The DL of AP-PP-DOX nanoparticles was 17.0 ± 1.7%, which was much higher than that of the nano drug delivery system based on PEG, of which the DL was usually under 5% [10,12]. This is because the number of carboxyl on the AP was much more than that on the PEG, thus allowing more DOX combined to the carrier.

### 2.3. MMP2-Dependent Drug Release

It is approved that the MMP2-sensitive peptide sequence (GPLGIAGQC) could be cleaved at the site between glycine (G) and isoleucine (I) by MMP2 when used as a linker in a synthetic peptide in the nanoparticles and drug conjugates [10,11]. Here, AP-PP-DOX could also be cleaved by the active human MMP2.

The drug release from the nanoparticles was investigated using the dialysis method under the simulated “sink” condition (0.1% Tween 80 in pH 7.4 PBS at 37 °C). The release profiles of the DOX-loaded nanoparticles after incubation with MMP2 were shown in Figure 5. The AP-PP-DOX nanoparticles showed a sustained drug release after incubation with MMP2 at 10 nM, 20 nM, and 40 nM. The cumulative releases of DOX were 74.5%, 71.8%, and 55.3% in 24 h, respectively. The drug release was only 8.7% after 24 h incubation in PBS. The nanoparticles showed a significant initial burst release (~45%, ~40%, and ~25% with MMP2 incubation at 10, 20, and 40 nM) at 1 h. Based on these results, we might predict that the nanoparticles would have a minimized drug leakage in circulation/normal tissues having a low MMP2 concentration. However, in the tumor tissue with high expression of MMP2, the nanoparticle would be decomposed, resulting in quick releasing of DOX and AP.

Although the loaded drugs were not 100% released in the in vitro condition, complete drug release would be expected once cell internalization due to cellular enzyme and low pH-mediated micelle disintegration/enhanced drug release [13].

### 2.4. Cellular Uptake of Nanoparticle

The cellular uptake of AP-PP-DOX nanoparticles with/without MMP2 preincubation was analyzed by LSCM and FACS Figure 6a,b). After 2 h of incubation, the free DOX showed high cellular uptake, which was attributed to the good free diffusion of DOX. The MMP2 preincubation could enhance the cellular uptake of DOX compared with that of the nanoparticles without MMP2 preincubation. This was because of the MMP2-triggered drug releasing, in agreement with the in vitro drug release result. Weak red fluorescence could be observed in nanoparticles without MMP2 preincubation, indicating that the nanoparticles could be internalized or probably the nanoparticles were cleaved by A549 cell-secreted MMP2.

### 2.5. Cytotoxicity of the Nanoparticles

The anticancer activity of the nanoparticles was tested with/without MMP2 incubation on cancer cell lines expressing different levels of MMP2, including A549 (MMP2 overexpressing) and MCF-7 (MMP2 low expressing). The cytotoxicity of AP-PP-DOX particles was increased in both A549 and MCF-7 in the presence of MMP2 compared with that of particles without MMP2 incubation (Figure 7a,b), which could be attributed to the MMP2 triggered quick DOX releasing. Notably, when incubated without MMP2, the nanoparticles showed higher cytotoxicity on A549 cells than that of MCF-7 cells. The different drug responses in these cells were mainly due to their different levels of extracellular MMP2. The data confirmed that the MMP2-sensitive drug releasing played a key role in cytotoxicity of the AP-PP-DOX particles.

### 2.6. Immunoregulatory Activity of Released AP Moiety by MMP2-Triggered Cleavage

Since the MMP2-sensitive peptide sequence (GPLGIAGQC) could be cleaved at the site between glycine (G) and isoleucine (I) by MMP2 when used as a linker, part of the peptide residue would be connected to the released AP moiety (AP-GPLG). Although the good immunomodulatory activity of AP had been proved in our previous studies [2,3,4,5], especially in promoting Th1-type immune response and inducing DC cell maturation [14,15], however, whether the peptide residue (-GPLG) would impact its activity need to be investigated further. As the results are shown in Figure 8a, the proliferation of mouse spleen cells was increased by AP-GPLG in a dose-dependent manner. The AP-GPLG also upregulated secretion of Th1-type cytokine IL-2 but downregulated secretion of Th2-type cytokine IL-10 (Figure 8b,c). The results were the same as that of AP itself we reported previously. Therefore, we would like to point out that the peptide residue (-GPLG) has no influence on the immunomodulatory activity of AP. The released AP moiety (AP-GPLG) would have the same immune behavior as AP itself.

Th1/Th2-type immune response plays an important role in regulating body immune activation/suppression. In tumor microenvironment, the Th1-type immune response was weakened while the Th2-type response was enhanced, resulting in weakening of the antitumor immunity [16]. The data indicated that the released AP moiety showed similar immunomodulatory activity as AP, which suggested that AP-PP-DOX nanoparticles have the potential to restore Th1/Th2 balance and improve the immunosuppressive state of tumor microenvironment.

## 3. Materials and Methods

### 3.1. Materials

*Angelica sinensis* polysaccharide (AP, MW50K) was made by our laboratory as reported previously [14]. The MMP-2 cleavable peptide (PP, NH_2_-GPLGIAGQC-SH, MW814) and human active MMP-2 enzyme (MW72K) were synthesized by Shanghai Xinhao Biological Technology Co., Ltd. (Shanghai, China). APMA (collagenase enzyme activator) was purchased from Shanghai Jiemei Gene Pharmaceutical Technology Co., Ltd. (Shanghai, China). Doxorubicin hydrochloride (DOX·HCl), 3-(Maleimido) propionic acid *N*-hydroxysuccinimide ester (SMP), *N*-(3-Dimethylaminopropyl)-*N*-ethylcarbodiimide hydrochloridecry-stalline (EDC), and *N*-Hydroxysuccinimide (NHS) were purchased from Sigma-Aldrich (Shanghai, China ). Maleic anhydride (MA) was purchased from Shanghai Chemical Plant (Shanghai, China). Triethylamine (TEA) was purchased from Tianjin Bodi Chemical Co., Ltd. (Tianjin, China). Lithium chloride (LiCl), dimethyl sulfoxide (DMSO), and dimethyl formamide (DMF) were purchased from Xi’an Chemical Reagent Factory (Shaanxi, China). Isopropyl alcohol was purchased from Dezhou Detian Chemical plant (Shandong, China). Cell culture medium DMEM, trypsin, and fetal bovine serum were purchased from Hangzhou Sijiqing Biological Engineering Materials Co., Ltd. (Hangzhou, China). MTT was purchased from Sai Cheng Biological Technology Co., LTD (Guangzhou, China). The human non-small cell lung cancer cells (A549) and breast cancer cells (MCF-7) were grown in complete growth media (DMEM supplemented with 1×PS and 10% FBS) at 37 °C in 5%CO_2_ air. A549 cells and MCF-7 cells were obtained from the Cell Bank of Type Culture Collection of the Chinese Academy of Sciences (Shanghai, China).

### 3.2. Synthesis and Characterization of AP-PP-DOX

AP-PP-DOX was synthesized according to the method reported [11,17] with some modifications. Three steps were involved (Figure 2). First, AP and maleic anhydride (MA) were reacted in LiCl/DMF (10 wt%) solvent system in the presence of triethylamine (TEA) under the protection of nitrogen at 60 °C for 20 h. The obtained AP-MA was precipitated in cold isopropyl alcohol followed by dialysis (MWCO 8000–14,000) against water for 48 h and lyophilization. Then, after activation by excess amount of the EDC/NHS, the AP-MA was conjugated with PP (1:1.2, molar ratio) in the carbonate buffer (pH 8.0) under nitrogen protection at 4 °C overnight. The product AP-PP was dialyzed (MWCO 8000–14,000) in water to remove the unreacted PP and lyophilized. The AP-MA and AP-PP were characterized by FT-IR and/or ^1^H-NMR using D_2_O as solvents. Second, the DOX was reacted with the SMP in DMF containing 0.3% of TEA in the dark at room temperature for 2 h. The red DOX-SMP was precipitated in ice diethyl ether and dried. The product DOX-SMP was characterized by FT-IR. Third, AP-PP with the terminal cysteine was reacted with the DOX-SMP in DMF containing 0.3% TEA under the protection of nitrogen in the dark at room temperature overnight. The obtained AP-PP-DOX was purified by dialysis (MWCO 8000–14,000) against water for 48 h and lyophilization. The final product AP-PP-DOX was characterized by 1H-NMR using D_2_O as solvents.

### 3.3. Nanoparticle Formation and Drug Loading

The AP-PP-DOX nanoparticles were prepared by self-assembly method. Briefly, the AP-PP-DOX conjugate (10 mg) was dissolved in 2 mL of DMSO and dialyzed (MWCO 8000–14,000) against water for 48 h. Then the solution was shaken ultrasonically for 2 min and filtered through a 0.45 μm syringe filter. The particle size and zeta potential of nanoparticles in PBS were measured by dynamic light scattering (DLS) on a Zetasizer Nano-ZS90 (Malven Panalytical) at 25 °C. The morphology of the nanoparticles was examined using transmission electron microscopy (TEM). The content of DOX in AP-PP-DOX was measured by ultraviolet spectrophotometer at 480 nm after decomposing by 0.1 mmol/L HCl. The drug loading (DL) was calculated by the following equations: DL (%) = (weight of loaded drug/weight of nanoparticles) × 100%(1)

### 3.4. In Vitro Drug Release

The in vitro drug release profile from the AP-PP-DOX nanoparticles was carried out by a dialysis method. Briefly, the AP-PP-DOX nanoparticles suspension was dialyzed (MWCO 8000–14,000) against PBS (pH 7.4) containing Tween 80 (0.1%, *w*/*v*) at 37 °C to maintain the sink condition with shaking. The released DOX was quantitated by ultraviolet spectrophotometer at 480 nm. To investigate the effect of the MMP2-mediated cleavage on the drug release rate, the nanoparticles were incubated with APMA-activated MMP2 (10–40 nMol/L) at 37 °C.

### 3.5. Cellular Uptake of Nanoparticles

The ability of mPEG-PMLA-DOX/DTX to enter cancer cells was studied by using confocal laser scanning microscopy and flow cytometry. Before the experiment, the A549 cells were seeded at a density of 1 × 10^5^ cells/well in dishes for 24 h. After being washed with PBS, the cells were incubated with nanoparticles in a serum-free medium for 2 h. Then, the cells were washed three times by the serum-free medium. For the fluorescence microscopy, the cells were fixed with 4% paraformaldehyde for 10 min. The cell nuclei were stained by DAPI for 5 min. The cells were observed on an Olympus FV 1000 fluorescence microscope system at a 60× magnification. For the flow cytometry analysis, the treated cells were harvested by trypsinization and centrifugation (600 g for 5 min). The collected cells were washed with PBS and resuspended in 200 μL PBS, followed by analysis on a BD FACSAria flow cytometer. The dead cells and cell debris were excluded from viable cells using the forward scatter (FSC) and side scatter (SSC). To study the effect of MMP2, the AP-PP-DOX nanoparticles were pretreated with human MMP2 (20 nmol/L) in pH 7.4 HBS at 37 °C overnight.

### 3.6. Cytotoxicity of the Nanoparticles

The A549 and MCF-7 cells were first seeded at 5 × 10^3^ cells/well in 96-well plates for 24 h. Then, the cells were incubated with the nanoparticles in the complete growth medium for 48 h. Finally, the cell viability was determined by the MTT assay.

### 3.7. Immunoregulatory Activity of Released AP Moiety by MMP2-Triggered Cleavage

The released AP moiety (AP-GPLG) triggered by MMP2 was obtained by means of dialysis method. Briefly, the AP-PP was first incubated with APMA-activated MMP2 (40 nMol/L) at 37 °C for 24 h. Then, the released AP-GPLG was obtained by dialysis (MWCO 8000–14,000) against water for 48 h and lyophilization.

BALB/c mice (6–7 weeks old) were obtained from the Experimental Animal Center, Air Force Medical University. Total splenocyte population was prepared from mice as described previously [14]. Spleen cells (10^5^ cells/well) were treated with AP-GPLG at different concentrations of 10–1000 μg/mL up to 72 h. The effect of AP-GPLG on proliferation of mouse spleen cells was detected by CCK-8 assay. The supernatants were collected and productions of IL-2 and IL-10 were measured using commercially available ELISA kit.

### 3.8. Data Analysis

Data were expressed as mean ± standard deviation (SD). Statistical differences among the control and different treatment groups were analyzed using one-way analysis of variance (ANOVA) and Fisher’s least significant difference (LSD) t-test. *p* < 0.05 was considered to be statistically significant.

## 4. Conclusions

In summary, the novel AP-PP-DOX nanoparticles with combination of chemotherapy and immunotherapy were successfully prepared, which had high drug-loading. The nanoparticles showed a good antitumor efficacy under the presence of MMP2. In the meantime, the released AP moiety still possessed the function of enhancing antitumor immune function, just the same as the polysaccharide itself. In this system, polysaccharides act as not only carriers but also effectors to generate a synergistic effect with chemotherapy drugs. Our results suggested that AP-PP-DOX nanoparticles might have great potential as a synergistic nanomaterial for tumor-targeted delivery of the anticancer drugs.

## Figures and Tables

**Figure 1 molecules-25-03096-f001:**
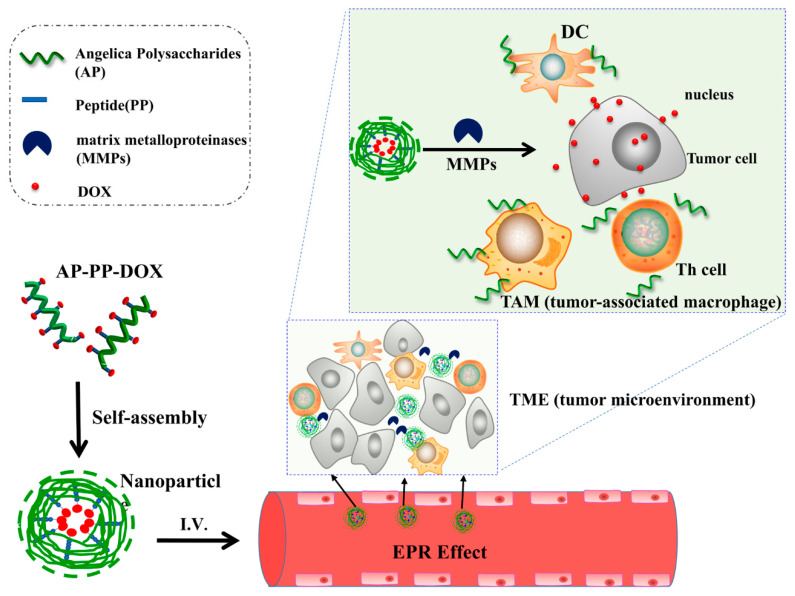
Proposed schematic diagram of AP-PP-DOX (*Angelica* polysaccharide-peptide-doxorubicin) nanoparticles for antitumor drug delivery.

**Figure 2 molecules-25-03096-f002:**
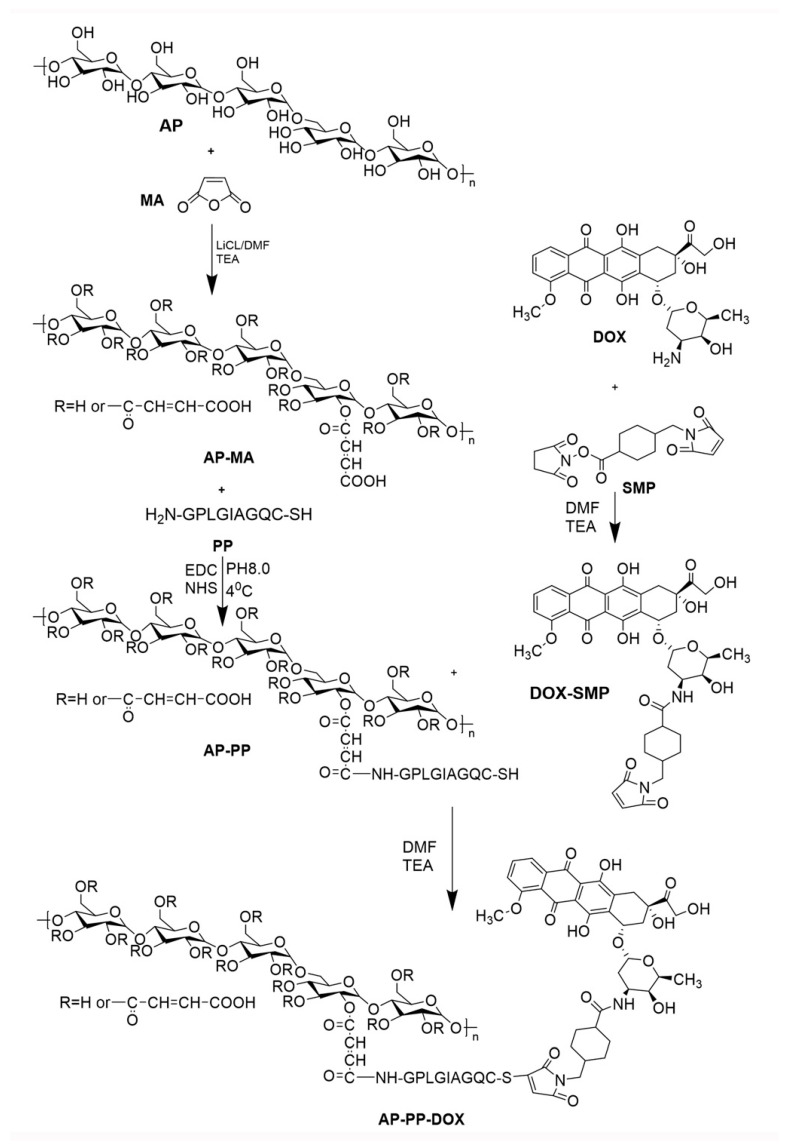
Synthetic scheme of AP-PP-DOX conjugates.

**Figure 3 molecules-25-03096-f003:**
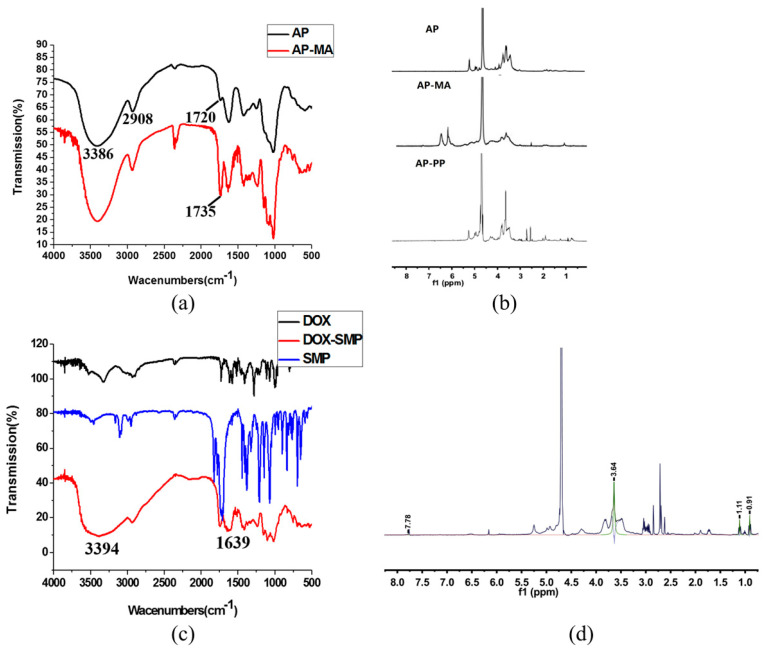
(**a**) FT-IR spectrum of AP and AP-maleic anhydride (MA). (**b**) ^1^H-NMR spectrum of AP, AP-MA, and AP-PP. (**c**) FT-IR spectrum of DOX, 3-(Maleimido) propionic acid *N*-hydroxysuccinimide ester (SMP), and DOX-SMP. (**d**) ^1^H-NMR spectrum of AP-PP-DOX.

**Figure 4 molecules-25-03096-f004:**
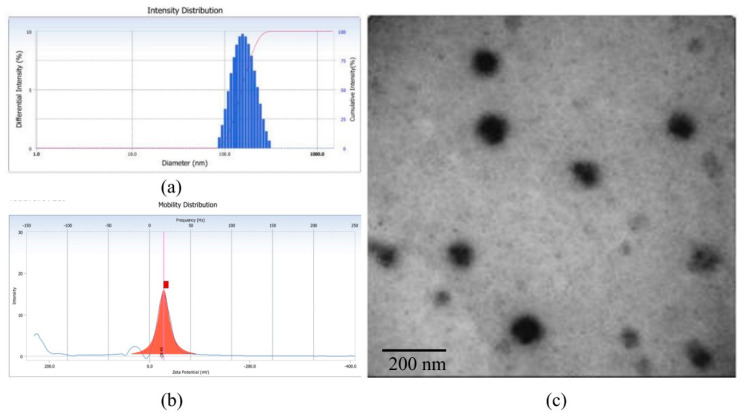
The size (**a**), zeta potential (**b**), and TEM image (**c**) of AP-PP-DOX nanoparticles.

**Figure 5 molecules-25-03096-f005:**
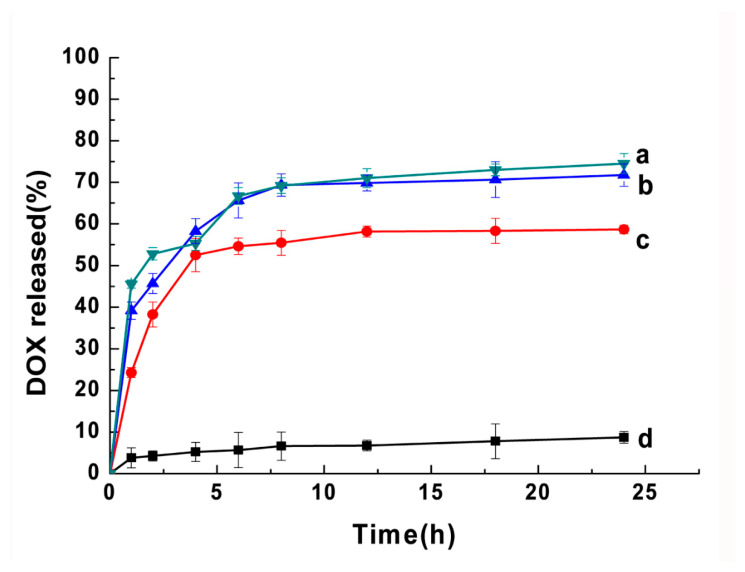
In vitro drug release profile of the nanoparticles. The AP-PP-DOX nanoparticles were incubated with PBS (**d**) or matrix metalloproteinase 2 (MMP2) at 10 nM (**c**), 20 nM (**b**), and 40 nM (**a**). Data were expressed as the mean ± SD.

**Figure 6 molecules-25-03096-f006:**
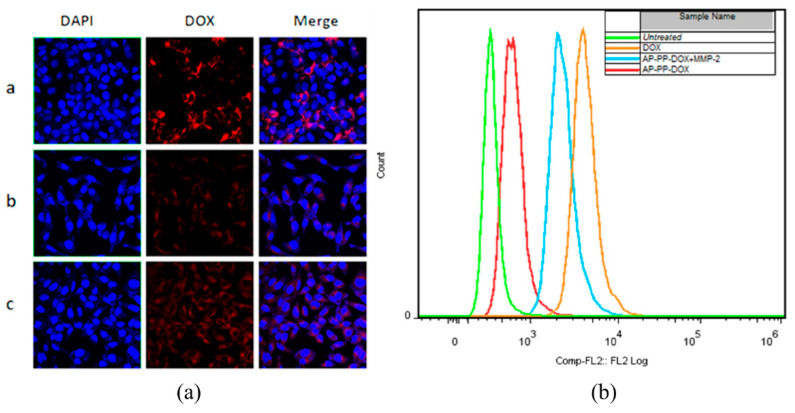
Cellular uptake of the nanoparticles after 2 h of incubation with the A549 cells, determined by LSCM (**a**) and FACS (**b**). a, Free DOX; b, AP-PP-DOX without MMP2 preincubation; c, AP-PP-DOX with MMP2 preincubation.

**Figure 7 molecules-25-03096-f007:**
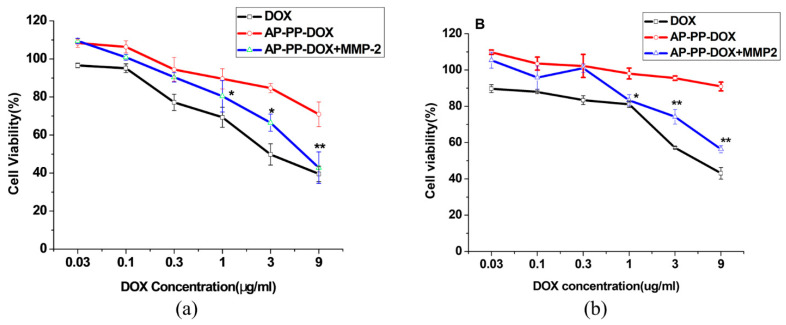
Cytotoxicity of AP-PP-DOX nanoparticles in cancer cells. (**a**) A549 (**b**) MCF-7. The cell viability assay was performed after 48 h with/without MMP2. Data were expressed as the mean ± SD. (*n* = 6, * *p* < 0.05, ** *p* < 0.01 vs. AP-PP-DOX).

**Figure 8 molecules-25-03096-f008:**
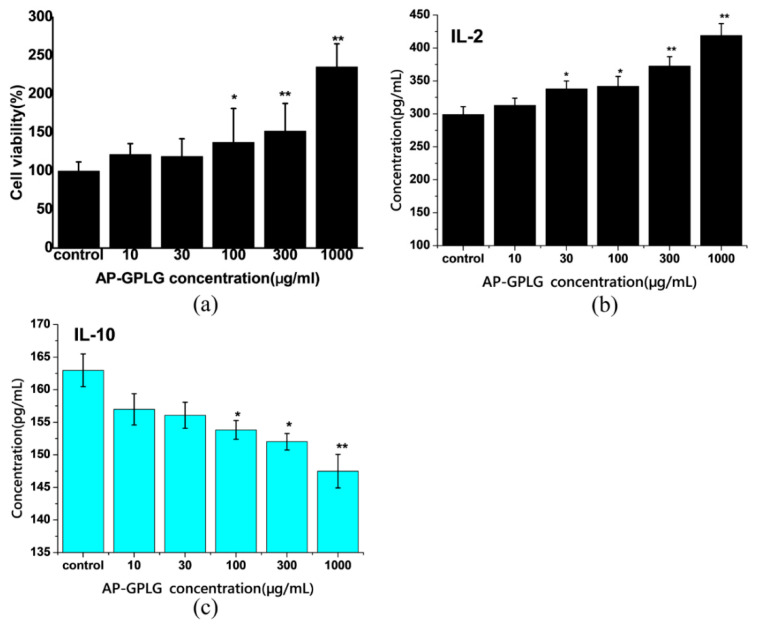
Effect of AP moiety (AP-GPLG) on proliferation of mouse spleen cells (**a**), IL-2 secretion (**b**), and IL-10 secretion (**c**). Data were expressed as the mean ± SD. (*n* = 6 in A, *n* = 3 in B and C, * *p* <0.05, ** *p* < 0.01 vs. control).

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
