# Peer review of "A Nano Drug Delivery System Based on Angelica sinensis Polysaccharide for Combination of Chemotherapy and Immunotherapy"

_molecules, 2020, doi:10.3390/molecules25133096_

Round 1
Reviewer 1 Report
This paper shared an innovative idea of a new drug delivery system by using the active polysaccharide as the carrier. The design is very interesting and the synthesis and chemical characterization are well completed. However, it is a pity that the authors failed to show any advantages brought by the new system in terms of improving solubility/bioavailability and enhancing targeted drug delivery. The drug was initially tested on only some in vitro models. It is suggested that the drug should be tested in vivo to show if it improves any drug properties. Publishing this paper at its current status will waste such a good idea.
Author Response
Thank you very much for your suggestions. In fact, the experiments in vitro models had preliminarily provided that the system, AP-PP-DOX, had good antitumor efficacy and enhanced antitumor immune function. In the next step, we plan to conduct experiments in vivo further and publish another article in Molecules.
Reviewer 2 Report
Abstract: The acronym of PP is not defined in the abstract (It seems to stand for peptide). Why don’t the authors use peptide instead of PP?
p.1, line 19: potential → zeta potential
p.2, Figure 1. The full name of MMP is not defined. So are TME and TAM.
Figure 4 caption: (c)
The abstract describes that the particle size was 139 ± 3 nm. However, its size was reported as 129.00 ± 3.32 nm in the text (p.5, line 95). Which one is correct?
The particle size data should include polydispersity index. A TEM image shown in Figure 4 suggests that the authors’ nanoparticles are heterogeneous in size.
p.9, line 233: The UV wavelength is not introduced. Any interfering effect by peptide on the UV absorbance of DOX?
Author Response
Reviewer #2:
Q1: Abstract: The acronym of PP is not defined in the abstract (It seems to stand for peptide). Why don’t the authors use peptide instead of PP?
A1: Thanks for your suggestion. The PP stands for peptide and we had added it in the abstract.
Q2: p.1, line 19: potential → zeta potential
A2: We had corrected it to “zeta potential”.
Q3: p.2, Figure 1. The full name of MMP is not defined. So are TME and TAM.
Figure 4 caption: (c)
A3: The full name of MMP (matrix metalloproteinases), TME (tumor microenvironment) and TAM (tumor-associated macrophage) had been defined in the Figure 1.
The mistake in Figure 4 had been corrected.
Q4: The abstract describes that the particle size was 139 ± 3 nm. However, its size was reported as 129.00 ± 3.32 nm in the text (p.5, line 95). Which one is correct?
A4: We are sorry for the mistake. The size was 129.00 ± 3.32 nm and we had corrected it.
Q5: The particle size data should include polydispersity index. A TEM image shown in Figure 4 suggests that the authors’ nanoparticles are heterogeneous in size.
A5: The polydispersity index was 0.119, which indicated the respectively uniform particle size.
Q6: p.9, line 233: The UV wavelength is not introduced. Any interfering effect by peptide on the UV absorbance of DOX?
A6: The UV wavelength used was 480 nm and was added in the revised article (p.9, line 226 and 233). Besides, there was no interfering effect by peptide on the UV absorbance of DOX and the UV absorption wavelength of AP-PP-DOX was 480 nm, just as the DOX.

Round 2
Reviewer 1 Report
The authors presented an interesting idea in this paper. It is publishable.